



# 1  A comprehensive biogeochemical record and annual flux estimates
# 2  for the Sabaki River (Kenya)

Trent R. Marwick[1], Fredrick Tamooh[2], Bernard Ogwoka[3], Alberto V. Borges[4], François Darchambeau[4],
and Steven Bouillon[1]
[1] Department of Earth and Environmental Sciences, KU Leuven, Leuven, 3001, Belgium
[2] Kenyatta University, Department of Zoological Sciences, Mombasa, Kenya.
[3] Kenya Wildlife Service, Mombasa, Kenya
[4] Unité d'Océanographie Chimique, Université de Liège, Liège, 4000, Belgium
*Correspondence to*: Trent R. Marwick (trent.marwick@gmail.com)

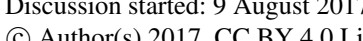



**Abstract.** Inland waters impart considerable influence on nutrient cycling and budget estimates across local, regional and
global scales, whilst anthropogenic pressures, such as rising populations and the appropriation of land and water resources,
are undoubtedly modulating the flux of carbon (C), nitrogen (N), and phosphorus (P) between terrestrial biomes to inland
waters, and the subsequent flux of these nutrients to the marine and atmospheric domains. Here, we present a two year
biogeochemical record (Oct. 2011 – Dec. 2013) at bi-weekly sampling resolution for the lower Sabaki River, Kenya, and
provide estimates for suspended sediment and nutrient export fluxes from the Athi-Galana-Sabaki (A-G-S) river basin under
pre-dam conditions, and in light of the approved construction of the Thwake Multi-purpose Dam on the Athi River. Erratic
seasonal variation was typical for most parameters, with generally poor correlation between discharge and material
concentrations and stable isotopic signatures of C ($\delta^{13}$C) and N ($\delta^{15}$N). Although high total suspended matter (TSM)
concentrations are reported here (up to ~3.8 g $L^{-1}$), peak concentrations of TSM rarely coincided with peak discharge. The
contribution of particulate organic C (POC) to the TSM pool indicates a wide bi-annual variation in suspended sediment load
from OC-poor (0.3%) to OC-rich (14.9%), with the highest %POC occurring when discharge is < 100 m$^3$ s$^{-1}$ and at lower
TSM concentrations. The consistent $^{15}$N enrichment of the PN pool compared to other river systems indicates anthropogenic
N-loading is a year-round driver of N export from the A-G-S basin. The Sabaki River was consistently oversaturated in
dissolved methane (CH$_4$; from 499% to 135,111%) and nitrous oxide (N$_2$O; 100% to 463%) relative to atmospheric
concentrations. We estimate export fluxes to the coastal zone of 4.0 Tg yr$^{-1}$, 70.6 Gg C yr$^{-1}$, 9.4 Gg N yr$^{-1}$, and 0.5 Gg P
yr$^{-1}$ for TSM, POC, and particulate forms of N (PN) and total P (TPP), respectively, and fluxes of 24.1 Gg C yr$^{-1}$, 6.6 Gg N
yr$^{-1}$, and 11.2 Gg P yr$^{-1}$ for dissolved forms of organic C (DOC), inorganic N (DIN), and phosphate (PO$_4^{3-}$). Wet season
flows (Oct. – Dec. and Mar. – May) carried > 80% of the total load for TSM (~86%), POC (~89%), DOC (~81%), PN
(~89%) and TPP (~82%), with > 50% of each fraction exported during the long wet season (Mar. – May). Our estimated
sediment yield of 85 Mg km$^{-2}$ yr$^{-1}$ is relatively low on the global scale and is considerably less than the recently reported
average sediment yield of ~630 Mg km$^{-2}$ yr$^{-1}$ for African river basins. Regardless, sediment and OC yields were all at least
equivalent or greater than reported yields for the neighbouring and flow-regulated Tana River. Rapid pulses of heavily $^{13}$C-
enriched POC coincided with peak concentrations of PN, ammonium, CH$_4$ and low dissolved oxygen saturation, lead to the
suggestion that large mammalian herbivores (e.g. hippopotami) may mediate the delivery of C$_4$ organic matter to the river
during the dry season. Given recent projections for increasing dissolved nutrient export from African rivers, as well as
planned flow regulation on the Athi River, these first estimates of material fluxes from the Sabaki River provide base-line
data for future research initiatives assessing anthropogenic perturbation of the A-G-S river basin.
**Copyright statement**
The authors agree with the licence and copyright agreement.





## 1 Introduction

The acknowledgement of the vital role inland waters play in carbon (C) cycling and budget estimates at local, regional and global scales has progressed steadily over the past three decades (e.g. Likens et al., 1981; Meybeck, 1982; Hedges et al., 1986; Kling et al., 1991; Cole et al., 1994; Ludwig et al., 1996; Richey et al., 2002; Cole et al., 2007; Battin et al., 2008; Tranvik et al., 2009; Bastviken et al., 2011; Raymond et al., 2013; Borges et al. 2015a), advancing to the state where individual components of the C budget of inland waters are included and parameterised within the Intergovernmental Panel on Climate Change (IPCC) budgeting of the global C cycle (see IPCC, 2013; also Ciais et al., 2013). For example, inland waters not only act as a conduit for the delivery of significant quantities of terrestrial organic C to the coastal zone and open ocean, they are typically sources of greenhouse gases (GHG's, $CO_2$, $CH_4$, $N_2O$) to the atmosphere, derived either from active heterotrophic metabolism remineralising a proportion of lateral inputs, through inputs from groundwaters and floodwaters which carry the products of mineralization in the terrestrial domain (Cole and Caraco, 2001a; Battin et al., 2009; Beaulieu et al., 2011; Raymond et al., 2013), or from wetlands (Abril et al. 2014; Borges et al. 2015a) with recent data compilations further elucidating the controls and drivers of GHG dynamics within the fluvial domain at regional and global scales (Borges et al., 2015a, Stanley et al., 2016, Marzadri et al., 2017). Additionally, a quantity of the lateral inputs may be buried within sedimentary deposits of reservoirs, lakes, floodplains and wetlands (Cole et al., 2007; Battin et al., 2008; Aufdenkampe et al., 2011). Anthropogenic pressures, such as land-use and land-use change, are undoubtedly modulating the quantities involved in each of these exchange fluxes (Regnier et al., 2013).

Given that recent reports assert a similar order of magnitude to the lateral C input to inland waters (~2.3 Pg C $yr^{-1}$) as that for global net ecosystem production (~2 Pg C $yr^{-1}$) (see Cole et al., 2007; Battin et al., 2009; Aufdenkampe et al., 2011; Ciais et al., 2013), the scarcity of the current empirical biogeochemical database for some regional inland waters is key to our inability to adequately resolve the role of this Earth System domain within broader regional and global C budgets (Raymond et al., 2013; Regnier et al., 2013). Although the spotlight has turned somewhat towards establishing a comprehensive reckoning of riverine C source variability and constraining C cycling within river basins, rather than solely quantifying the transport fluxes from inland waters to the coastal zone (Bouillon et al., 2012), there remain important inland water systems or regions lacking long-term, riverine biogeochemical datasets built upon high frequency sampling initiatives capable of providing reliable transport flux estimates. Tropical and sub-tropical Africa is one region where such datasets are scarce (e.g. Coynel et al., 2005; Borges et al. 2015a). On the global scale, the tropics and subtropics are considered of particular importance regarding the transport of sediments and C (Ludwig et al., 1996; Schlünz and Schneider, 2000; Moore et al., 2011), with a recent compilation of African sediment yield (hereafter, SY) data highlighting the paucity of observations relative to other continental regions (Vanmaercke et al., 2014). Also, the inland waters of the tropics and subtropics are suggested to have elevated evasion rates of $CO_2$ to the atmosphere in comparison to temperate and boreal inland waters (Aufdenkampe et al., 2011; Raymond et al., 2013; Borges et al. 2015a,b), and the same has been asserted for global $CH_4$ flux from tropical rivers and lakes (Bastviken et al., 2011; Borges et al. 2015a,b). Hence, given their reported significance as a





source of GHGs to the atmosphere, an increased focus on the inland water biogeochemistry of the tropics is merited (Regnier
et al., 2013; Stanley et al., 2016), particularly for data-scarce river basins of Africa, given these regions contribute some of
the largest uncertainty to global C budgets (Ciais et al., 2011).
Over the preceding decade, momentum has gathered towards a broader understanding of the nutrient cycling within sub-
Saharan inland water ecosystems (e.g. Coynel et al., 2005; Brunet et al., 2009; Abrantes et al., 2013; Zurbrügg et al., 2013;
Bouillon et al., 2014). Yet, Africa has experienced the highest annual population growth rate over the preceding 60 years
(~2.51%, 1950 – 2013; see United Nations, 2013), a position it is expected to hold for the remainder of the 21st century
(United Nations, 2013). Coupling the increasing population with forecasted climate change scenarios, land-use changes
including deforestation and expanding agricultural practises, hydrological flow regulation through dam and reservoir
construction and water abstraction, as well as increased exploitation of natural resources for food, fuel and wood products,
will shift the dynamics of lateral nutrient inputs to inland waters of Africa, as well as the balance between transport and in-
situ processing of these terrestrial subsidies, and consequently the regional C and nutrient balance of Africa (Hamilton, 2010;
Yasin et al., 2010; Ciais et al., 2011; Valentini et al., 2014). Hence, continued effort in characterising the biogeochemistry of
African inland waters is paramount for developing robust regional and global nutrient budgets, but also to provide a working
baseline for assessing future climate and land-use impacts on the nutrient fluxes to and from inland waters of Africa.
The potential perturbation of the biogeochemistry of tropical inland waters by climate and land-use change (Hamilton,
2010), and those of Africa specifically (Yasin et al., 2010), has received some attention. Given a projected warming of a ~2 –
4.5 ˚C toward the end of the 21st century within the tropics (Meehl et al., 2007; Buontempo et al., 2015) and in East Africa
specifically (Buontempo et al., 2015; Dosio and Panitz, 2016), important shifts are predicted involving: (i) aquatic thermal
regime, influencing rates of in-situ microbe-mediated biogeochemical processes, (ii) hydrological regimes of discharge and
floodplain inundation, and (iii) freshwater-saltwater gradients, altering biogeochemical processing as rivers approach the
coastal zone. Additionally, Yasin et al. (2010) estimate that the load of all dissolved and particulate forms of C, N, and P in
African river basins have increased in the period 1970 – 2000, and further increases are predicted for all dissolved fractions
of N and P between 2000 – 2050, although C fractions and particulate forms of N and P are modelled to decrease. Predicted
decreases of particulate loads are linked to the net effect of climate change and reservoir construction, which alter hydrology,
nutrient retention and sediment carrying capacity of rivers (Yasin et al., 2010), and which store ~25% of annual sediment
load carried over the African landmass (Syvitski et al., 2005), while the increasing dissolved nutrient loads are related to the
rising population, as well as increased per capita gross domestic product (GDP) and meat consumption, with these factors
driving up the terrestrial inputs of manure, fertiliser and sewage derived N and P (Yasin et al., 2010).
British settlement brought European land-use practises to the Kenyan highlands early in the 20th century, triggering severe
soil erosion in, and elevated sediment fluxes from, the Athi-Galana-Sabaki (A-G-S) River basin (Champion, 1933; Fleitmann
et al. 2007). These terrigenous sediments have had a significant impact on the environment surrounding the outflow of the
Sabaki River in the Indian Ocean, for example, by increasing coral stress (van Katwijk et al., 1993) and spreading seagrass
beds on local reef complexes, as well as siltation and infilling of the Sabaki estuary and the rapid progradation of nearby



shorelines (Giesen and van de Kerkhof, 1984). In order to alleviate regional water scarcity, construction of reservoirs on the
Athi River have been under consideration for decades, the implementation of which could modify the magnitude of sediment
delivery to the coastal zone (van Katwijk et al., 1993) as previously observed in the neighbouring Tana River (Finn, 1983;
Tamooh et al., 2012).
Here, we present a 2-year biogeochemical record at fortnightly resolution for the riverine end-member of the A-G-S system,
and in light of the planned construction of the Thwake Multi-purpose Dam (currently awaiting tender approval, see
http://www.afdb.org/projects-and-operations/project-portfolio/project/p-ke-e00-008/), we provide estimates for sediment and
nutrient export rates from the A-G-S system whilst still under pre-dam conditions.
**2 Materials and methods**
**2.1 Study area**
The Athi-Galana-Sabaki River basin is the second largest drainage basin (~46600 km$^2$) in Kenya. The headwaters are located
in central and south-east Kenya, in the vicinity of Nairobi city (Fig. 1), draining agricultural areas (predominantly tea and
coffee plantations) which provide the livelihood of 70% of the regional population (Kithiia, 1997). Industrial activities and
informal settlements dominate land-use around Nairobi, with livestock and small-scale irrigation activities also present
downstream. The basin landcover is dominated by grasslands biomes (~65%) rich in $C_4$ species (Fig. 1), with agriculture
accounting for ~15% and the region of Nairobi <1%. Forest biomes dominated by $C_3$ vegetation are isolated to higher
altitude regions in the basin headwaters, as well as in the coastal region where the Sabaki River discharges to the Indian
ocean at Malindi Bay (Fig. 1).
Precipitation ranges between 800 and 1200 mm yr$^{-1}$ in the highly populated central highlands surrounding Nairobi, to 400–
800 mm yr$^{-1}$ in the less populated, lower altitude, and semi-arid south-east of Kenya. Two dry seasons (January–February,
hereafter JF; June–September, hereafter JJAS) intersperse a long (March–May, hereafter MAM) and short (October–
December, hereafter OND) wet season. Only during the MAM and OND periods does monthly precipitation exceed potential
evaporation-transpiration within the basin (Fig. 1), and accordingly the annual hydrograph displays bimodal discharge, with
an average flow rate of 49 m$^3$ s$^{-1}$ between 1957–1979 (Fleitmann et al. 2007). Dry season flow rates as low as 0.5 m$^3$ s$^{-1}$
compare to peak wet season flow rates of up to 5000 m$^3$ s$^{-1}$ (Delft Hydraulics, 1970; Fleitmann et al., 2007). Oscillations
between El Niño and La Niña conditions have a strong influence on the decadal patterns of river discharge, where extended
severe drought is broken by intense and destructive flooding (Mogaka et al., 2006). The pre-1960 sediment flux of 0.06 Tg
yr$^{-1}$ is dwarfed by modern day flux estimates of 5.7 and 14.3 Tg yr$^{-1}$ (Van Katwijk et al., 1993; Kitheka, 2013), with the
rapid increase in sediment flux over the preceding half-century attributed to a combination of intensified land use practices,
the highly variable climatic conditions and extremely erosive native soils. More detailed information regarding basin settings
may be found in Marwick et al. (2014a).





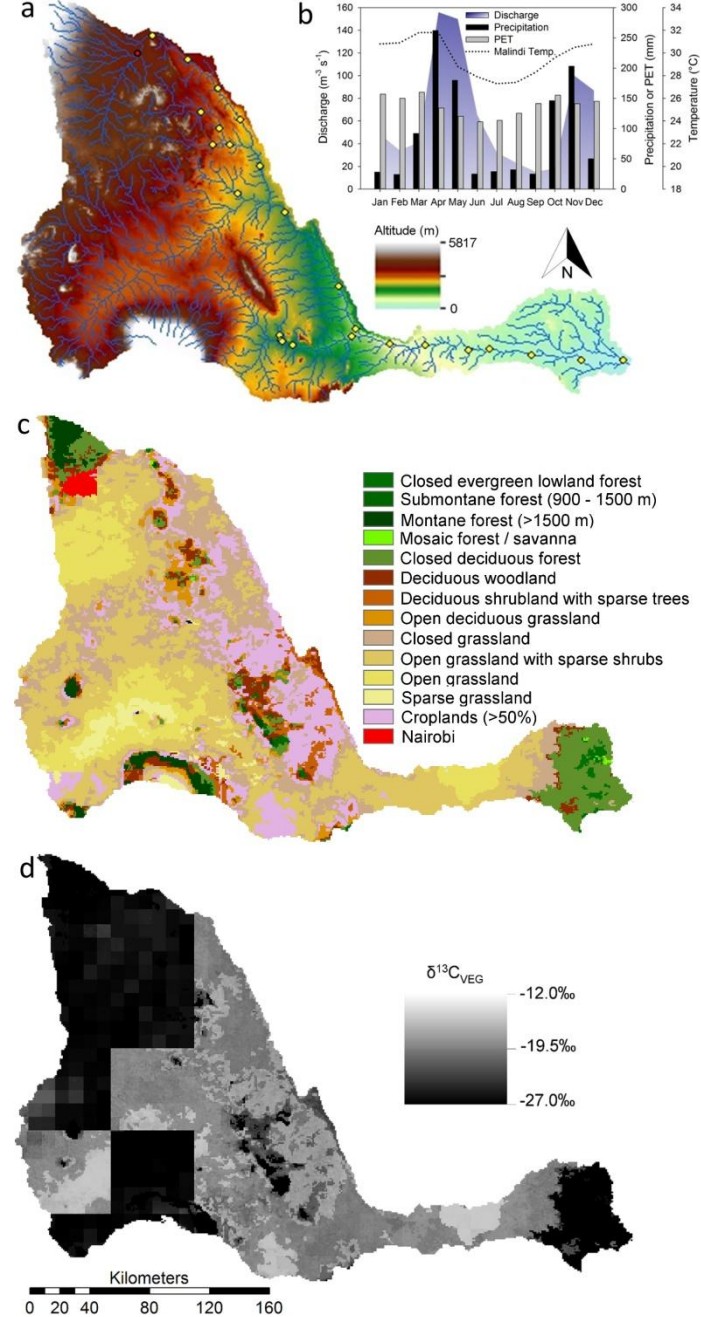

**Figure 1. The Athi-Galana-Sabaki River basin: (a) digital elevation model, (b) mean monthly variation of hydrological and climate parameters including discharge at the outlet (shaded area; data from 1959–1977), precipitation (black bar) (from Fleitmann et al., 2007), potential evapotranspiration (PET; grey box), and the maximum air temperature in Malindi (A-G-S outlet; dotted black line), (c) GLC2000 vegetation biomes (Mayaux et al., 2004), and (c) Crop corrected vegetation *isoscape* (extracted from Still and Powell (2010)). The yellow dots in (a) mark the site locations from Marwick et al. (2014a). Data presented here was collected at the most eastern sampling locality (site S20 from Marwick et al. (2014)), while our discharge estimates were calculated from data collected at the adjacent site directly west (site S19 from Marwick et al. (2014)).**



## 2.2 Sampling and analytical techniques

Physico-chemical parameters and biogeochemistry of the Sabaki River were monitored bi-weekly (i.e. fortnightly) approximately 2 km upstream of Sabaki Bridge (approximately 5 km upstream of the river outlet to Malindi Bay) for the period October 2011 to December 2013. Water temperature, conductivity, dissolved oxygen ($O_2$) and pH were measured in situ with a YSI ProPlus multimeter, whereby the $O_2$ and pH probes were calibrated on each day of data collection using water saturated air and United States National Bureau of Standards buffer solutions (4 and 7), respectively. Samples for dissolved gases ($CH_4$, $N_2O$) and the stable isotope composition of dissolved inorganic C ($\delta^{13}C_{DIC}$) were collected from mid-stream at ~0.5 m depth with a custom-made sampling bottle consisting of an inverted 1L polycarbonate bottle with the bottom removed, and ~0.5 m of tubing attached in the screw cap (Abril et al. 2007). 12 mL exetainer vials (for $\delta^{13}C_{DIC}$) and 50 mL serum bottles (for $CH_4$ and $N_2O$) were filled from water flowing from the outlet tubing, poisoned with $HgCl_2$, and capped without headspace. Approximately 2000 mL of water was collected 0.5 m below the water surface for other particulate and dissolved variables, and filtration and sample preservation was performed in the field within 2 h of sampling. Samples for total suspended matter (TSM) were obtained by filtering 60-250 mL of water on pre-combusted (4 h at 500°C) and pre-weighed glass fibre filters (47mm GF/F, 0.7 μm nominal pore size), and dried in ambient air during the fieldwork. Samples for determination of particulate organic C (POC), particulate nitrogen (PN) and C isotope composition of POC ($\delta^{13}C_{POC}$) were collected by filtering 40-60 mL of water on pre-combusted 25 mm GF/F filters (0.7 μm nominal pore size). The filtrate from the TSM filtrations was further filtered on 0.2 μm polyethersulfone syringe filters (Sartorius, 16532-Q) for total alkalinity (TA), DOC and $\delta^{13}C_{DOC}$ (8-40 mL glass vials with Polytetrafluoroethylene coated septa). TA was analysed by automated electro-titration on 50 mL samples with 0.1 mol $L^{-1}$ HCl as titrant (reproducibility estimated as typically better than ± 3 μmol $kg^{-1}$ based on replicate analyses).

For the analysis of $\delta^{13}C_{DIC}$, a 2 ml helium (He) headspace was created, and $H_3PO_4$ was added to convert all DIC species to $CO_2$. After overnight equilibration, part of the headspace was injected into the He stream of an elemental analyser – isotope ratio mass spectrometer (EA-IRMS, ThermoFinnigan Flash HT and ThermoFinnigan DeltaV Advantage) for $\delta^{13}C$ measurements. The obtained $\delta^{13}C$ data were corrected for the isotopic equilibration between gaseous and dissolved $CO_2$ as described in Gillikin and Bouillon (2007), and measurements were calibrated with certified reference materials LSVEC and either NBS-19 or IAEA-CO-1. Concentrations of $CH_4$ and $N_2O$ were determined via the headspace equilibration technique (20 mL $N_2$ headspace in 50 mL serum bottles) and measured by gas chromatography (GC, Weiss 1981) with flame ionization detection (GC-FID) and electron capture detection (GC-ECD) with a SRI 8610C GC-FID-ECD calibrated with $CH_4$:$CO_2$:$N_2O$:$N_2$ mixtures (Air Liquide Belgium) of 1, 10 and 30 ppm $CH_4$ and of 0.2, 2.0 and 6.0 ppm $N_2O$, and using the solubility coefficients of Yamamoto et al. (1976) for $CH_4$ and Weiss and Price (1980) for $N_2O$.

25 mm filters for POC, PN and $\delta^{13}C_{POC}$ were decarbonated with HCl fumes for 4 h, re-dried and packed in Ag cups. POC, PN, and $\delta^{13}C_{POC}$ were determined on the abovementioned EA-IRMS using the thermal conductivity detector (TCD) signal of the EA to quantify POC and PN, and by monitoring m/z 44, 45, and 46 on the IRMS. An internally calibrated acetanilide and





sucrose (IAEA-C6) were used to calibrate the $\delta^{13}C_{POC}$ data and quantify POC and PN, after taking filter blanks into account.
Reproducibility of $\delta^{13}C_{POC}$ measurements was better than ±0.2 ‰. Samples for DOC and $\delta^{13}C_{DOC}$ were analysed either on a
Thermo HiperTOC IRMS (Bouillon et al. 2006), or with an Aurora1030 TOC analyser (OI Analytical) coupled to a Delta V
Advantage IRMS. Typical reproducibility observed in duplicate samples was in most cases <±5 % for DOC, and ±0.2 ‰ for
$\delta^{13}C_{DOC}$.
Our dataset for $CH_4$ and $N_2O$ has been used in a continental-scale data synthesis in Borges et al. (2015a), but are discussed
here in more detail.

## 2.3 Discharge estimates

Historical discharge observations and daily gauge height data for the sampling period was provided by the Water Resource
Management Authority (WRMA), Machakos, Kenya. Due to the poor resolution of discharge and gauge data at the Sabaki
Bridge north of Malindi (gauge # 3HA06) over the monitoring period, the finer fidelity record from the Baricho station
(gauge # 3HA13) was used, situated approximately 50 km upstream of our biogeochemical monitoring station (i.e. site S20
from basin-wide sampling campaigns, see Marwick et al. 2014a). With discharge measurements from 2006 and 2007 (*n* =
11), care of WRMA, we developed a rating curve to calculate daily discharge from available gauge data (Fig. 2a). As seen in
Fig. 2a, the limited and poor spread of discharge measurements results in extrapolation for gauge heights < 1m and > 3m.
Although Kenyan rivers have been suggested to export up to 80% of annual sediment load during pulse discharge events
over few days (*Dunne,* 1979), the timeframe of these event pulses is typically short-lived relative to more mundane flow
conditions, and at heights for example < 3m (which account for ~97% of gauge data) we have reasonable confidence that the
rating curve reflects in-situ conditions. Given the general positive correlation between discharge and sediment concentration,
and disregarding possible hysteresis in discharge-sediment flux dynamics (which have been shown for the neighbouring
Tana River), we suspect the greatest error in our discharge estimates is when gauge height exceeds 3 m.
The Baricho gauge height dataset contains a 2 month period of no measurement (1st of February to 31st March, 2013). For
this period, the daily discharge was estimated as the average discharge for that day over the previous 10 years (2003 − 2012).

## 2.4 Suspended sediment and C, N, and P flux estimates

Annual flux estimates for suspended sediments and the various riverine fractions of particulate and dissolved C,
N and P were calculated with the discharge data above. We interpolated linearly between the concentrations
measured on consecutive sampling dates in order to establish concentrations for every day of the study period.
The daily concentrations were then multiplied by daily discharge and summed over the study period to establish
annual flux estimates.



## 3 Results

### 3.1 Discharge

All data (excluding results for $NH_4^+$, $NO_3^-$ and $PO_4^{3-}$) are presented for the period between October 2011 and September 2013, encompassing two full seasons each of short wet (Oct. – Dec.; OND), short dry (Jan. – Feb.; JF), long wet (Mar. – May; MAM) and long dry (Jun. – Sep.; JJAS). Over the monitoring period, daily discharge (Fig. 2b; see also Supplementary Materials, Table 1) varied between 13 $m^3 s^{-1}$ and 2032 $m^3 s^{-1}$, with mean and median flow rates of 139 $m^3 s^{-1}$ and 51 $m^3 s^{-1}$, respectively, compared to the average flow rate of 73 $m^3 s^{-1}$ reported by Kitheka (2013) for 2001 – 2003 and noted as a relatively wet period. The average annual discharge throughout the monitoring period totalled ~4.4 $km^3$, considerably less than the ~10.7 $km^3$ used by Mayorga et al. (2010) and approximately double that reported by Kitheka (2013) (~2.3 $km^3$) for the period 2001 – 2003. There was negligible inter-annual variation of total discharge for the monitoring period. Discharge during the wet seasons (MAM + OND) accounted for 82% and 79% of annual discharge for 2011 – 2012 and 2012 – 2013, respectively, while 59% and 51% of annual discharge occurred during the upper 10% of daily flows for the same periods.

Sampling of $NH_4^+$, $NO_3^-$ and $PO_4^{3-}$ was conducted over a different timeframe to the rest of the data presented here. The range in daily discharge over this time period (21st Dec. 2012 to 20th Dec. 2013) reflects the ranges reported above, although the mean flow rate was somewhat elevated (169 $m^3 s^{-1}$). Total annual discharge was 5.3 $km^3$, with between 83% of total annual discharge occurring during the wet seasons.

Throughout the Results and Discussion we use discharge values of $\leq 68$ $m^3 s^{-1}$ and $\geq 152$ $m^3 s^{-1}$ when referring to low and high flow (hereafter LF and HF) conditions respectively, corresponding to the maximum value for the upper 80% of daily dry season flows and minimum value for the upper 30% of daily wet season flows.

### 3.2 Physico-chemical parameters

Water temperature varied from 24.1˚C to 33.9˚C (average ± 1 SD = 29.8 ± 2.0˚C), with considerable variability intra- and inter-seasonally. The coolest temperatures occurred at the end of the MAM wet season and during the JJAS dry season. pH varied widely across the sampling period (range = 4.6 to 10.1) yet maintained an average of 7.1 ± 1.1. Most basic conditions were typically observed during lower flow periods of the dry seasons. %$O_2$ saturation ranged between 23.3% and 130.0%, with least saturated conditions observed during the JJAS dry season of 2013. There was no clear relationship between discharge and conductivity, with the latter's range varying sporadically over the sampling period from 113.0 µS cm$^{-1}$ to 1080.0 µS cm$^{-1}$ (average = 487.1 ± 254.5 µS cm$^{-1}$). Total alkalinity (TA) varied over an order of magnitude (0.475 to 4.964 mmol kg$^{-1}$) with an average of 2.438 ± 0.872 mmol kg$^{-1}$. There was poor correlation between discharge and TA, with





observed peaks scattered across the hydrograph, suggesting a simple two source scenario of baseflow and high flow dilution
is inadequate to explain the seasonal variability for the A-G-S system. All data for physico-chemical parameters and those
outlined below are presented in Table 1 of the Supplementary Materials.

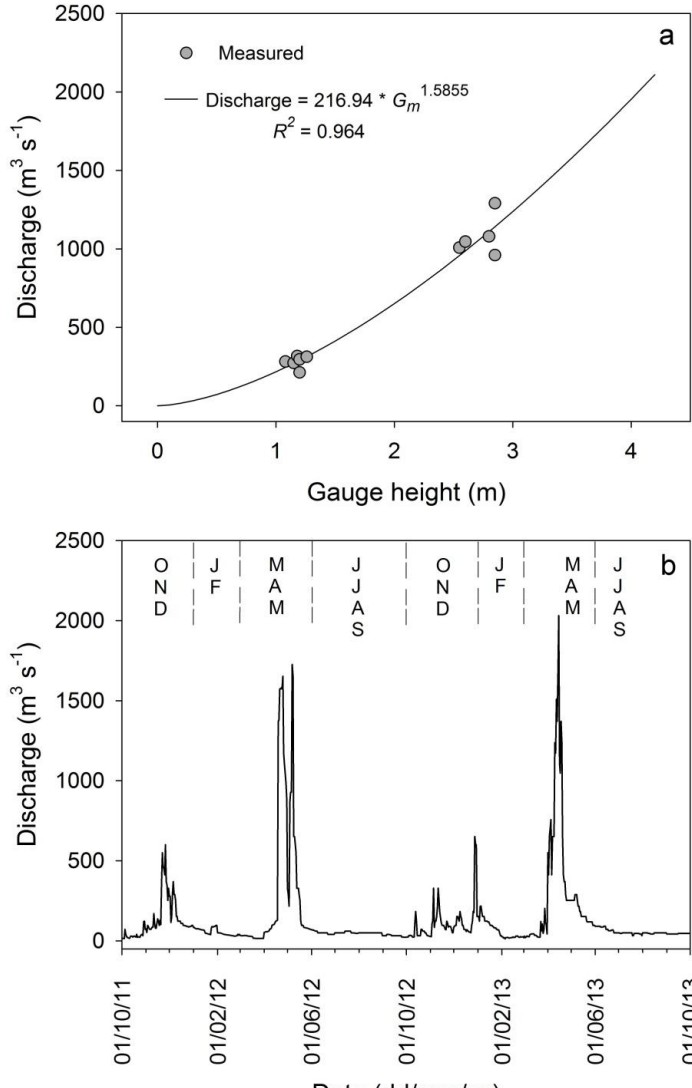

**Figure 2. (a) Discharge rating curve for the Sabaki River at the Baricho gauge station (3HA13). (b) Calculated daily discharge for**
**the two year monitoring period. Note, one anomalous gauge reading on the 12/11/2012 provides an upper discharge estimate of**
**41332 $m^3 s^{-1}$, over a magnitude larger than the next highest daily discharge estimate (3441 $m^3 s^{-1}$). Given the discharge estimates**
**on the preceding (11/11/2012) and following days (13/11/2012) were 312 and 218 $m^3 s^{-1}$, respectively, and also reported historical**
**maximum daily discharge of ~5000 $m^3 s^{-1}$ (Delft Hydraulics, 1970), we linearly interpolated the gauge data for the 12/11/2012 from**
**the values of adjacent days thereby lowering the discharge estimate for this date to 249 $m^3 s^{-1}$. The curve in (a) was developed**
**from the limited dataset (n = 11) of recent discharge measurements (2006 – 2007; grey circles) on the Sabaki River at Baricho.**
**(data supplied by WRMA, Machakos)**



### 3.3 Bulk concentrations

The concentrations of TSM, POC, particulate N (PN) and total particulate phosphorus (TPP) are shown in Fig. 3, as well as the stable isotope composition of POC and PN, with most variables showing complex variation across the hydrological year. The Sabaki River exported TSM varying in concentration from 50.0 to 3796.7 mg $L^{-1}$ (Fig. 3a), containing POC at concentrations between 3.5 and 74.6 mg $L^{-1}$ (Fig. 3b). The lower and upper TSM and POC concentrations were associated with the JJAS (dry) and OND (wet) periods of 2011 respectively. The contribution of POC to the TSM pool (hereafter, %POC) indicates a wide bi-annual variation in suspended sediment load from OC-poor (0.3%) to OC-rich (14.9%), with the highest %POC occurring when discharge is < 100 $m^3 s^{-1}$ (Fig. 4a) and at lower TSM concentrations (Fig. 4b). The large range for the C stable isotope ($\delta^{13}C$) of the POC pool ($\delta^{13}C_{POC}$; −23.3‰ to −14.5‰) displayed complex temporal patterns with no obvious trends across seasons nor with discharge (Fig. 3b). Particulate N ranged in concentration from 0.3 to 9.4 mg $L^{-1}$ (Fig. 3c), while the ratio of POC to PN (as a weight:weight ratio; hereafter, POC:PN) varied from 6.6 to 17.4, with an average value of 9.4 ± 1.7 ($n = 42$). The N stable isotope composition ($\delta^{15}N$) of PN ($\delta^{15}N_{PN}$) showed considerable fluctuation (from −3.1 to +15.9‰; Fig. 3c), with the most $^{15}N$- enriched PN recorded at the beginning of the OND period of 2011 – 2012 and during the JJAS period of 2012 – 2013. The TPP load showed complex temporal variability (Fig. 3d), with concentrations ranging between 61.2 and 256.1 μg $L^{-1}$ and displayed negligible correlation with discharge. Although TPP generally rose during (or slightly preceding) peak discharge, the highest values were recorded under LF conditions during the 2012 – 2013 JJAS period.

The dissolved organic C (DOC) concentration fluctuated from 3.3 to 9.3 mg $L^{-1}$ (Fig. 5a), with lowest and highest concentrations observed during the JJAS and MAM periods of 2013, respectively. The highest DOC concentrations were regularly observed in the weeks following wet season peak discharge. The contribution of DOC to the total OC (TOC) pool ranged between 15% and 68% (accounting for 20% and 32% of annual TOC export during 2011 – 2012 and 2012 – 2013 respectively) with no clear seasonal trend. Akin to the $\delta^{13}C_{POC}$, the $\delta^{13}C$ composition of the DOC pool ($\delta^{13}C_{DOC}$) displayed complex variability over a large range (−29.3‰ to −17.9‰) with no obvious relationship with either seasonality or discharge (Fig. 5a). On average, the DOC was more depleted in $^{13}C$ than concurrent POC samples ($\delta^{13}C_{POC} - \delta^{13}C_{DOC} = 2.8 ± 2.9‰$, n = 40).

The $\delta^{13}C$ composition of the DIC pool ($\delta^{13}C_{DIC}$) shifted between −12.4‰ and −3.2‰ and also shows a complex pattern across the hydrograph (Fig. 5b), though the DIC pool was generally more enriched in $^{13}C$ during LF periods and more $^{13}C$-depleted over the wet seasons.

The concentration range for $NH_4^+$, $NO_3^-$, and $PO_4^{3-}$ over the 1-yr period were 7.1 to 309.6 μmol $L^{-1}$, <0.1 to 506.9 μmol $L^{-1}$, and 1.1 to 322.6 μmol $L^{-1}$ respectively (Fig. 6). No strong seasonal pattern is apparent in the dissolved inorganic N fractions (Figs. 6b and 6c), although peak concentrations generally occur at below average discharge conditions (i.e. when Q < 169 $m^3 s^{-1}$ then the average (± 1 SD) DIN concentration is 172.2 ± 140.1 μmol $L^{-1}$ (n = 20), whereas when Q ≥ 169 m3 $s^{-1}$ then the average (± 1 SD) DIN concentration is 59.6 ± 26.3 μmol $L^{-1}$ (n = 5)). The concentration of $PO_4^{3-}$ displayed an erratic pattern





1. over the course of the year (Fig. 6d). Concentrations were highly variable at below average flow conditions (i.e. when $Q <$

2. 169 $m^3$ $s^{-1}$ the average ($\pm$ 1 SD) $PO_4^{3-}$ concentration is 105.7 $\pm$ 97.2 µmol $L^{-1}$ (n = 20)), whereas concentrations became

3. comparatively low during above average discharge (i.e. when $Q \geq 169$ m3 $s^{-1}$ then the average ($\pm$ 1 SD) $PO_4^{3-}$ concentration

4. is 34.8 $\pm$ 31.0 µmol $L^{-1}$ (n = 5)).

5. The river was consistently oversaturated in dissolved $CH_4$ relative to the atmosphere (from 499% to 135,111%) with a

6. concentration range between 10 and 2,838 nmol $L^{-1}$ (Fig. 7a). Although $CH_4$ peaks occurred in both dry and wet season, the

7. largest annual peaks occur at the end of the JJAS dry period. Concentrations of dissolved $N_2O$ (Fig. 7b) varied from 5.9 and

8. 26.6 nmol $L^{-1}$, corresponding to oversaturation of 100% to 463% relative to atmospheric concentrations. $N_2O$ concentrations

9. were highest during the OND period of 2011 – 2012, and otherwise showed maximum concentrations preceding peak

10. discharge during the MAM period of each year.

11. **3.4 Annual flux and yield of particulate and dissolved fractions**

12. Annual material flux estimates to the coastal zone for TSM and various C, N, and P fractions are provided in Table 1.

13. Briefly, our data suggest a mean flux of 4.0 Tg $yr^{-1}$, 70.6 Gg C $yr^{-1}$ and 24.1 Gg C $yr^{-1}$ for TSM, POC and DOC

14. respectively, corresponding to mean annual %POC of 1.8%, and mean annual contribution of DOC to the TOC pool

15. (hereafter %DOC) of 26%. Bi-annually, wet season (OND, MAM) flows carried >80% of the total load for TSM (~86%),

16. POC (~89%) and DOC (~81%), with the MAM period accounting for > 50% of TSM, POC and DOC annual export.

17. Estimates of mean annual flux of PN and TPP were 7.5 Gg and 0.5 Gg respectively, and > 80% of bi-annual export of PN

18. (~89%) and TPP (~82%) occurred during the wet seasons, with > 50% of the annual flux occurring over the MAM period.

19. Annual dissolved nutrient flux estimates (Table 1) were 2.3 Gg, 4.3 Gg and 11.2 Gg for $NH_4^+$, $NO_3^-$ and $PO_4^{3-}$ respectively.

20. Approximately 75% of $NH_4^+$ export occurred during the wet seasons, whereas only 66% of $NO_3^-$ export occurred over the

21. same period. Approximately 79% of annual $PO_4^{3-}$ export took place during the wet seasons, with a greater proportion

22. exported over the OND wet season (45%) than the MAM wet season.

23. Various surface area estimates are reported for the A-G-S basin, ranging from 40000 $km^2$ (Giesen and van de Kerkhof, 1984;

24. van Katwijk et al., 1993), to ~70000 $km^2$ (Fleitmann et al., 2007; Kitheka, 2013), and up to 117000 $km^2$ by Mayorga et al.

25. (2010). Using ArcGIS 10.1 and the river basins of Africa output of Lehner et al. (2006) (http://hydrosheds.cr.usgs.gov), we

26. estimate the A-G-S basin covers an area of ~46750 $km^2$.

27. Taking the above basin area estimate and the flux values detailed above, we estimate mean annual yields of 84.6 Mg $km^{-2}$,

28. 1.51 Mg C $km^{-2}$ and 0.52 Mg C $km^{-2}$ for TSM, POC and DOC respectively (Table 1). Conservative mean annual yields for

29. PN and TPP were 161 kg N $km^{-2}$ and 11 kg P $km^{-2}$, while those of the dissolved fractions over the single hydrological year

30. were 49 kg N $km^{-2}$, 93 kg N $km^{-2}$ and 239 kg P $km^{-2}$ for $NH_4^+$, $NO_3^-$ and $PO_4^{3-}$, respectively (see also Supplementary

31. Material, Table 2).





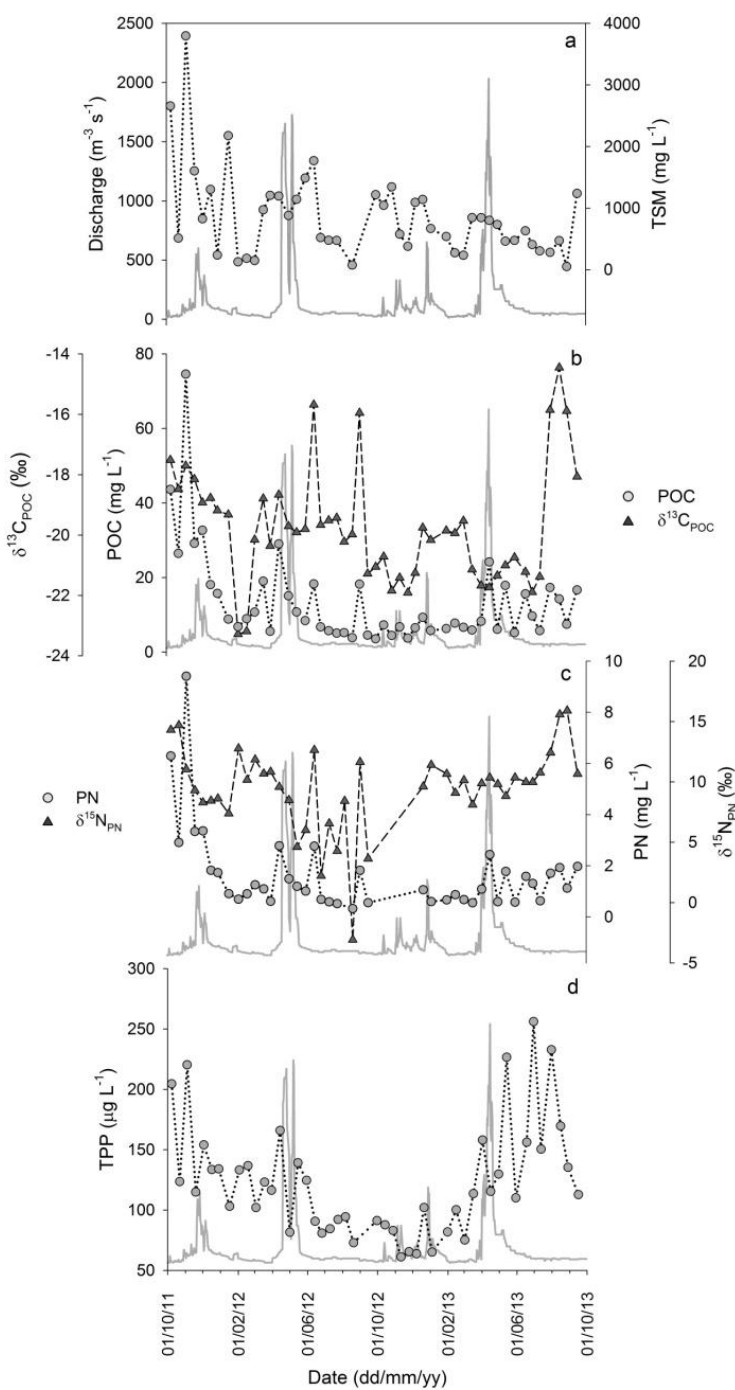

**Figure 3. Discharge (solid grey line) and two years of monitoring the (a) total suspended matter concentration, the concentration and stable isotope signature of (b) particulate organic carbon and (c) particulate nitrogen, and the concentration of (d) total particulate phosphorus in the Sabaki River. In all figures grey circles represent bulk concentrations and dark triangles represent stable isotope signatures.**







**Figure 4. The relationship between the % contribution of particulate organic carbon to the total suspended load and (a) discharge, and (b) total suspended matter. The dashed line in (a) marks discharge of 100 m³ s⁻¹, as cited in-text.**





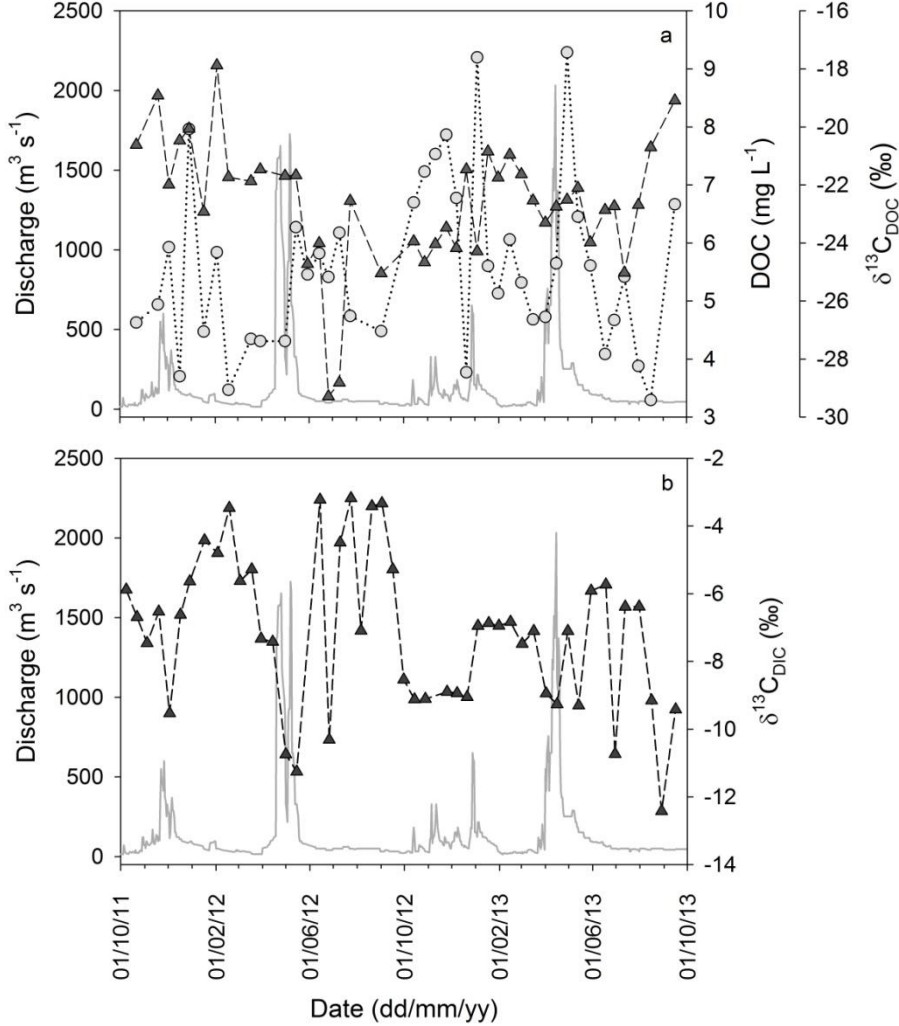

2 **Figure 5. Discharge and two years of monitoring the dissolved (a) organic carbon concentration and carbon stable isotope**
3 **signature, and (b) the carbon stable isotope signature of dissolved inorganic carbon in the Sabaki River. Grey circles represent**
4 **bulk concentrations, with dark triangles for all stable isotope signatures.**



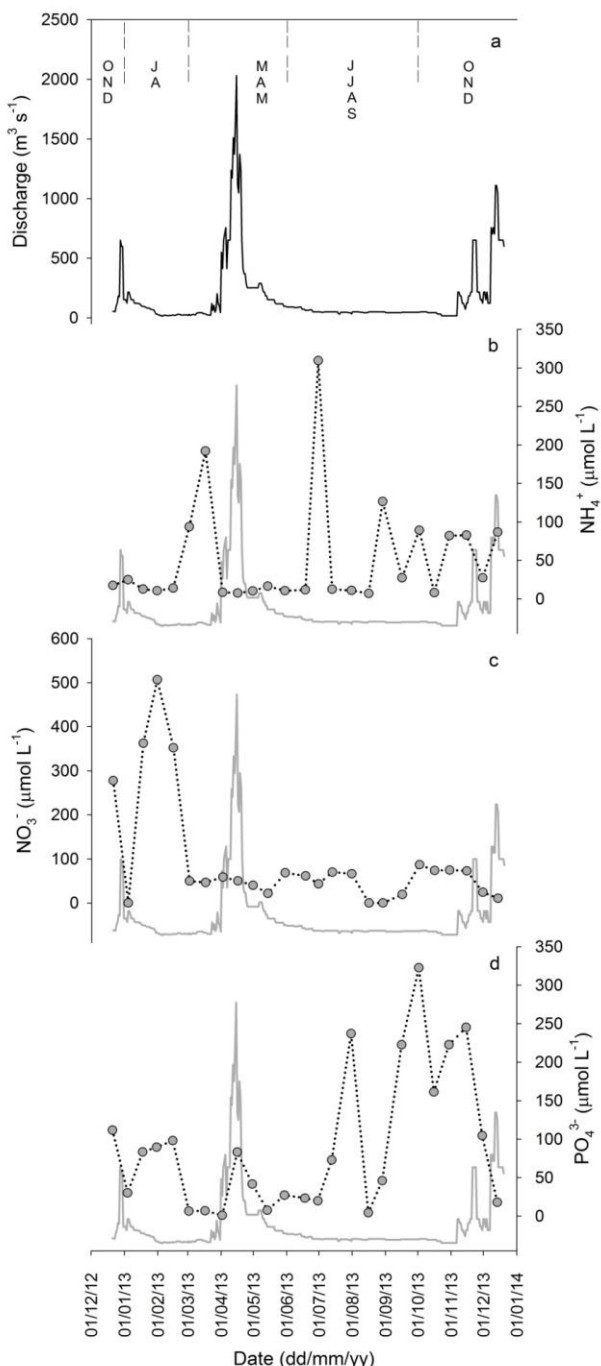

**Figure 6. (a) Daily discharge rates and one year of monitoring the concentration of dissolved (b) ammonium, (c) nitrate and (d)**
**phosphate in the Sabaki River. In figures (b) – (d) grey circles represent bulk concentrations.**



**Figure 7. Two years of monitoring concentrations of dissolved (a) methane and (b) nitrous oxide. Grey circles represent riverine gas concentrations.**





**Table 1. Summary of annual fluxes, element ratios, and annual yields for the Athi-Galana-Sabaki basin from data reported here**
**and from the NEWS2 export model (see Mayorga et al., 2010), as well as data for 2012 and 2013 from the neighbouring Tana**
**River basin at Garsen (Geeraert et al., in review).**

| | A-G-S | A-G-S (NEWS2) | Tana |
|---|---|---|---|
| *Flux* | | | |
| Basin area (km$^2$) | 46750 | 117230 | 81700 |
| Discharge (km$^3$ yr$^{-1}$) | 4.39$^a$ | 10.75 | 4.32 - 4.71 |
| Discharge (km$^3$ yr$^{-1}$) | 5.32$^b$ | | |
| | | (Tg yr$^{-1}$) | |
| TSM | 4.0 | 38.8 | 4.1 - 4.9 |
| | | (Gg yr$^{-1}$) | |
| POC | 70.6 | 205.3 | 113 - 157 |
| DOC | 24.1 | 49.5 | 11 - 14 |
| PN | 9.4 | 16.4 | |
| TPP | 0.5 | 9.6 | |
| DIN | 6.6 | 7.4 | |
| PO$_4$$^{3-}$ | 11.2 | 0.9 | |
| %POC (of TSM) | 1.8 | 0.5 | |
| POC:PN | 8.7 | 12.5 | |
| %DOC (of TOC) | 25.5 | 23.0 | |
| *Yield* | | (Mg km$^{-2}$ yr$^{-1}$) | |
| TSM | 84.6 | 330.7 | 50.2 - 60.0 |
| POC | 1.51 | 1.75 | 1.38 - 1.92 |
| DOC | 0.52 | 0.42 | 0.13 - 0.17 |
| | | (kg km$^{-2}$ yr$^{-1}$) | |
| PN | 161 | 140 | |
| TPP | 11 | 82 | |
| DIN | 142 | 63 | |
| PO$_4$$^{3-}$ | 239 | 8 | |

$^a$ All fractions except dissolved N and P: hydrological years 1st October 2011 to 30th September
2012 and 1st October 2012 to 30th September 2013.
$^b$ Dissolved N and P only: hydrological year 21st December 2012 to 14th December 2013.
**4 Discussion**
Although previous studies provide estimates of annual suspended sediment fluxes at the Sabaki outlet as well as annual yield
estimates for the A-G-S basin (Watermeyer, 1981; Munyao et al., 2003; Kitheka, 2013), their primary research focus lay
elsewhere, and none provide the comprehensive biogeochemical record at a comparable temporal scale as presented here.
The following discussion revolves around the main objectives of our study, including: (i) the quantification of annual





suspended matter, C, N and P fluxes and sediment yield, (ii) characterising the sources of particulate and dissolved fractions
of C and N, and (iii) to provide indications to the water-atmosphere transfer of important greenhouse gases ($CH_4$ and $N_2O$) at
the outlet of the Sabaki River. We conclude with consideration of the future anthropogenic impacts in the A-G-S basin and
the consequences for material fluxes from the Sabaki River to the coastal zone.
**4.1 Material fluxes, annual yields and their origin**
To the best of our knowledge, and excluding suspended matter, the estimates provided in Table 1 are the first quantifications
of material fluxes from the A-G-S system. A suspended sediment flux of ~7.5 to 14.3 Tg yr$^{-1}$ is commonly cited for the A-
G-S system (Watermeyer et al., 1981; van Katwijk et al., 1993; Fleitmann et al., 2007), which is approximately 2- to 3.5-fold
greater than our conservative TSM flux estimate of ~4.0 Tg yr$^{-1}$. A more recent estimate from Kitheka (2013) for the period
2001 – 2003 (5.7 Tg yr$^{-1}$) is still greater than, though more comparable to, our own estimate above. Whereas we employed
year-round bi-weekly monitoring and extrapolated fluxes from daily gauge height readings, Kitheka (2013) measured
concurrent discharge and suspended matter concentrations at monthly to bi-weekly periodicity. The relative coarseness of
sampling interval employed by Kitheka (2013), in combination with their acknowledgement that peak sediment flux often
occurs prior to peak discharge i.e. sediment exhaustion effect (Rovira and Batalla, 2006; Oeurng et al., 2011; Tamooh et al.,
2014), may pre-empt accurate extrapolation of the annual sediment flux from their limited dataset. For example, in order to
accurately estimate fluxes in systems with an irregular hydrograph, such as the neighbouring Tana River (which experiences
similar climatic conditions and annual hydrograph pattern to the A-G-S basin), monitoring at a recurrence interval of < 7
days has been recommended (Tamooh et al., 2014), also implying that the flux estimates presented here may be improved
with a more refined sampling frequency.
If we normalise the basin area of ~70000 km$^2$ reported by Fleitmann et al. (2007) and Kitheka (2013) to the value reported
here (~46750 km$^2$), and subsequently recalculate their SY from their riverine sediment flux values, we find our SY of ~85
Mg km$^{-2}$ yr$^{-1}$ is considerably lower than the 160 to 306 Mg km$^{-2}$ yr$^{-1}$ recalculated from Fleitmann et al. (2007) and the 122
Mg km$^{-2}$ yr$^{-1}$ from Kitheka (2013).
Some have reported that prior to 1960 the suspended sediment load of the A-G-S basin was ~58 Gg yr$^{-1}$ (Watermeyer et al.,
1981; Van Katwijk et al., 1993), which is equivalent to a SY of ~1 Mg km$^{-2}$ yr$^{-1}$. Although indeed the A-G-S basin has been
disturbed by anthropogenic practises since European arrival, this value needs to be met with some scepticism, as it represents
an approximately 85-fold increase in annual soil loss over the preceding 50 years. In the neighbouring Tana River basin,
Tamooh et al. (2014) estimated annual suspended sediment yields between 46 and 48 Mg km$^{-2}$ at ~150 km from the river
mouth (basin area of 66500 km$^2$). More recently, higher resolution dataset of Geeraert et al. (in review; see Table 1) for the
Tana River at Garsen (~70 km from the river mouth, basin area of 81700 km$^2$) estimated a suspended SY of 50 – 60 Mg
km$^{-2}$, indicating that the relatively smaller A-G-S basin exports a comparable quantity of sediment annually to the coastal
zone as that discharged from the much larger (and heavily regulated) Tana River basin.



The SY reported here is low compared to the global average of 190 Mg km$^{-2}$ yr$^{-1}$ (Milliman and Farnsworth, 2011) and
considerably less than the average of 634 Mg km$^{-2}$ yr$^{-1}$ for the African continent recently reported by Vanmaercke et al.
(2014). This may be somewhat surprising given the typically concentrated suspended sediment loads observed over the
monitoring period (mean (± 1 SD) = 865 ± 712 mg L$^{-1}$; median = 700 mg L$^{-1}$), but can be explained by the fact all TSM
concentrations > 1500 mg L$^{-1}$ were observed at below HF discharge rates (i.e. < 152 m$^{-3}$ s$^{-1}$; see Fig. 3a). All the same, our
SY estimate is over 3-fold greater than the average pre-dam SY of 25 Mg km$^{-2}$ yr$^{-1}$ from the Congo, Nile, Niger, Zambezi
and Orange rivers (draining > 40% of the African landmass) (Milliman and Farnsworth, 2011). Sediment yield estimates
from other arid tropical basins of Africa (e.g. Gambia, Limpopo, Niger, and Senegal rivers) are significantly lower (between
3 to 18 Mg km$^{-2}$ yr$^{-1}$; Milliman and Farnsworth, 2011), although reported yields of 94 Mg km$^{-2}$ yr$^{-1}$ from the Rufiji
(Tanzania) and 88 Mg km$^{-2}$ yr$^{-1}$ from the Ayensu (Ghana), both arid tropical basins, are equivalent to what was observed in
the A-G-S basin.
The annual POC yield (1.5 Mg C km$^{-2}$) from the A-G-S basin is equivalent to the global average of 1.6 Mg C km$^{-2}$ (Ludwig
et al., 1996), though almost triple the estimate of 0.6 Mg C km$^{-2}$ by Tamooh et al. (2014) at their most downstream site on
the neighbouring Tana River, and over seven-fold greater than the 0.2 Mg C km$^{-2}$ reported from the largely pristine, wooded
savannah dominated Oubangui River (Bouillon et al., 2014), the 2$^{nd}$ largest tributary to the Congo River. The over-riding
influence of sewage inputs on the biogeochemistry of the A-G-S basin has been previously brought to attention by Marwick
et al. (2014a), partially through investigation of the $\delta^{15}$N composition of the PN pool. The average $\delta^{15}$N$_{PN}$ recorded across the
monitoring period here was 9.5 ± 3.5‰ ($n$ = 43), which sits above the 75$^{th}$ percentile of measurements within other African
basins (see Marwick et al. (2014a), Fig. 10 therein), and reflects the range of $\delta^{15}$N signatures of NH$_4^+$ (+7‰ to +12‰; Sebilo
et al., 2006) and NO$_3^-$ (+8‰ to +22‰; Aravena et al., 1993; Widory et al., 2005) sourced from raw waste discharge. As
highlighted earlier, around 50% percent of Nairobi's population of 3 million live in slums with inadequate waste
management facilities which leads to increasing water quality issues (Dafe, 2009; Kithiia and Wambua, 2010), providing an
evident explanation for the POC-loaded sediment flux from the A-G-S basin in comparison to other African river basins.
The annual DOC yield from the A-G-S basin (0.5 Mg C km$^{-2}$) is markedly lower than the global mean of 1.9 Mg C km$^{-2}$
(Ludwig et al., 1996). The DOC yield is within the range of 0.1 to 0.6 Mg C km$^{-2}$ reported for the Tana River (Tamooh et
al., 2014), consistent with the global observation of low DOC concentrations in rivers of semi-arid regions (Spitzy and
Leenheer, 1991), and also falls between observations in tropical savannah basins of ~0.3 Mg C km$^{-2}$ for the Gambia River
(Lesack et al., 1984) and ~0.9 Mg C km$^{-2}$ for the Paraguay River (Hamilton et al., 1997). Tamooh et al. (2014) attributed the
low DOC yield in the Tana basin to low soil OC content (average of 3.5 ± 3.9% OC) as well as high temperatures in the
lower basin (Tamooh et al., 2012 and 2014). Surface soils (0 – 5 cm) in the A-G-S basin were of low OC content also,
ranging between 0.4 to 8.9% OC with an average value of 2.0 ± 1.9% ($n$ = 19; own unpublished data), although due to site
selection, samples were not gathered from the relatively OC-rich soils of the upper A-G-S basin (see
http://www.ciesin.columbia.edu/afsis/mapclient/ and overlay 'Soil Organic Carbon Mean – Depth 0 – 5 cm').



In contrast to some other $C_4$-rich tropical and sub-tropical river basins, the POC load in the Sabaki River (average $\delta^{13}C$ =
$-19.7 \pm 1.9‰$) is marginally enriched in $^{13}C$ compared to the basin-wide bulk vegetation $\delta^{13}C$ value of $-21.0‰$, as estimated
from the crop corrected vegetation *isoscape* of Africa in Still and Powell (2010) (Fig. 1c). For example, in the $C_4$-dominated
Betsiboka River basin of Madagascar, a consistent underrepresentation of $C_4$-derived C in riverine OC pools was reported by
Marwick et al. (2014b), with similar observations (particularly during dry season) within the Congo (Mariotti et al., 1991;
Bouillon et al., 2012) and Amazon (Bird et al., 1992) basins and in rivers of Australia (Bird and Pousai, 1997) and Cameroon
(Bird et al., 1994 and 1998), and is typically attributed to a greater portion of riverine OC sourced from the neighbouring $C_3$-
rich riparian zone relative to more remote $C_4$ dominated landscapes (i.e. grassland/savannah). Under this scenario, the $C_4$-
derived riverine OC component generally peaks during the wet season in response to the increased mobilisation of surface
and sub-surface OC stocks from more distant $C_4$-rich sources. At the outlet of the A-G-S basin, on the other hand, not only
was POC more enriched in $^{13}C$ (peak value of $-14.5‰$) than values recorded in the neighbouring Tana basin ($-19.5‰$;
Tamooh et al. (2014)) or the $C_4$-dominated Betsiboka basin ($-16.2‰$; see Marwick et al. (2014b)), but these $^{13}C$-enriched
POC loads occurred during consecutive JJAS periods (i.e. long dry season), and therefore, an alternative mechanism to the
*riparian zone effect* outlined above is required to explain these dry season observations. One possibility is herbivore-
mediated inputs of $C_4$-derived OM to riverine OC pools, such as from livestock or large native African mammals, as has
been reported for Lake Naivasha (Grey and Harper, 2002) and the Mara River in Kenya (Masese et al., 2015). The combined
Tsavo West and Tsavo East National Parks, accounting for approximately 4% of the total surface area of Kenya, are
dissected by the Galana River downstream of the confluence of the Tsavo with the Athi River. These national parks contain
large populations of mammalian herbivores (Ngene et al., 2011), including elephants and buffalo (Supplementary Figure 1a
and 1b, respectively), which graze on the $C_4$ savannah grasses and gravitate towards perennial water sources, such as the
Galana River, during the dry season. More importantly, hippopotami (Supplementary Figure 1c) graze within the $C_4$-rich
savannas by night and excrete partially decomposed OM to the river during the day. Grey and Harper (2002) estimated the
total quantity of excrement for the Lake Naivasha hippopotami population to be ~5.8 Gg yr$^{-1}$ (~500 individuals), assuming a
consumption of 40 kg of biomass and a measured maximum wet weight of 8 kg of excrement on land per individual per
night, with the remainder excreted to the lake during the day. This equates to approximately ~12 Mg yr$^{-1}$ per hippopotamus,
and using the mean excrement compositions from Grey and Harper (2002) of 37% carbon and 1.5% nitrogen, results in
hippopotamus-mediated delivery of ~740 kg C yr$^{-1}$ and ~30 kg N yr$^{-1}$. To a lesser extent, additional terrestrial subsidies
would be supplied by livestock using the river as a water source (Supplementary Figure 1d). Aerial census results from 2011
identified ~80 hippopotami within the combined Tsavo East (i.e. Athi and Galana rivers) and Tsavo West (i.e. Tsavo River)
National Parks, considerably less than the ~4000 reported from the Masai Mara National Reserve where the research of
Masese et al. (2015) was conducted. Supplementary Figure 1 highlights the high density of other large mammals
congregating around the Athi and Galana rivers, and though a smaller proportion of their total excrement will be released
directly to the river relative to hippopotami, the combined quantity may be a significant contribution to the riverine OC pool
under low flow conditions. Hence, it is reasonable to assume these herbivores deliver significant quantities of $C_4$-derived



OM to inland waterways, especially during the dry season when other local water sources are depleted, with this being a time
when the inputs may be particularly noticeable in riverine $\delta^{13}C_{POC}$ signatures, as the contribution from other allochthonous
sources would be minimised (especially $C_4$-derived OM, see Marwick et al. (2014b)) due to lower terrestrial runoff rates.
The correlation between minor peaks in bulk POC and $^{13}C$ enriched $\delta^{13}C_{POC}$ signatures during the JJAS period of 2012
supports this suggestion, when without a simultaneous increase in discharge, a short pulse of $C_4$-derived OC is observed in
the Sabaki River.
The findings from the basin-wide campaigns reported in Marwick et al. (2014a) led to the suggestion that the concentration
of DIN in export from the A-G-S basin likely peaks during the wet season, due to the significant processing and removal of
DIN in the upper- to mid-basin during the dry season and which resulted in significantly lower DIN concentration at the
monitoring station (i.e. site S20 from Marwick et al. (2014a)) relative to wet season observations. Our higher-resolution
dataset, however, suggests a more complex relationship between DIN concentrations, seasonality, and discharge, given that
peak DIN concentrations were also observed during low flow conditions (Fig. 6b and 6c). In particular, a prominent $NH_4^+$
peak during the JJAS dry season of 2013 occurred in conjunction with peaks in POC and PN, and might be attributed to in-
situ processing of the dry season organic matter inputs from large herbivores in the lower basin, as outlined above. Similarly,
a prominent peak in $NO_3^-$ was observed during the JF dry season, for which no clear explanation exists. Despite this, our flux
estimates suggest that the annual DIN and PN export predominantly occurs during the wet seasons as a result of the elevated
discharge conditions, and with the consistent enrichment of the PN pool in $^{15}N$ (Fig. 3d) relative to the $\delta^{15}N$ composition of
biologically fixed N (i.e. ~0‰ to +2‰), supports the analysis of Marwick et al. (2014a) that anthropogenic inputs impart
significant influence on the cycling of N in the A-G-S basin and the export budget of N from the Sabaki River to the coastal
zone.
The Global Nutrient Export from Watersheds 2 (NEWS2; see Mayorga et al. (2010)) provides flux and yield estimates for
TSM and particulate and dissolved fractions of organic and inorganic forms of C, N, and P for > 6000 river basins through
hybrid empirical and conceptual based models relying on single and multiple linear regressions and single-regression
relationships. Comparatively, our flux estimates are in general considerably lower than the NEWS2 estimates (Table 1),
except for the dissolved $PO_4^{3-}$ pool. There are at least three likely explanations for these over estimates. Firstly, the basin
area used in NEWS2 calculations is 2.5-fold greater than our estimate, and given the flux estimates of Mayorga et al. (2010)
are also a function of basin area, it is understandable there will be considerable over-estimation by the model. Secondly, the
TSM sub-model is grounded in datasets of observed conditions (generally not impacted by extensive flow regulation) and
independent factors including precipitation, a relief index, dominant lithology, wetland rice and marginal grassland extent,
whereas the export of particulate forms of C, N, and P are reliant on empirical relationships between contents of TSM and
POC (Ludwig et al., 1996) and POC and PN (Ittekkot and Zhang, 1989), and a relationship for particulate phosphorus export
based on POC load developed by Beusen et al. (2005). We suggest these relationships may not extrapolate well to a basin so
severely impacted by anthropogenic inputs as the A-G-S system. Thirdly, export of dissolved fractions is built upon an
empirical dataset from 131 global river basins, though this includes only nine African basins, compared to 45 basins for



North America and 36 basins for Europe for example, and hence the relationships developed from these datasets will be
biased towards conditions observed in these regions and not necessarily reflective of African systems. Additionally, the
NEWS2 model only takes into account contributions from sewage when areas are connected to sewage systems (i.e. point
source inputs), which is not the case for 1.5 million residents of Nairobi, and may explain the major underestimation of the
dissolved $PO_4^{3-}$ flux.
**4.2 Greenhouse gases**
The combination of high frequency sampling and long-term monitoring of dissolved $CH_4$ and $N_2O$ concentrations in the
rivers of Africa remain scarce (Borges et al., 2015a). The average and median concentrations of $CH_4$ in the Sabaki River
($483 \pm 530$ nmol $L^{-1}$ and $311$ nmol $L^{-1}$, respectively; $n = 50$) often exceeded observations in other rivers of Africa, including
the mid-and lower-Tana River ($54 - 387$ nmol $L^{-1}$; Bouillon et al. (2009)), the Comoé, Bia and Tanoé rivers of Ivory Coast
($48 - 870$ nmol $L^{-1}$; Koné et al. (2010)), and the Oubangui River of Central African Republic ($74 - 280$ nmol $L^{-1}$; Bouillon
et al. (2012)). On a seasonal basis, $CH_4$ concentrations tended to rise and fall with discharge (Fig. 7a), opposite to
observations in the Oubangui and Ivory Coast rivers where highest concentrations are observed during low flow periods and
decrease as discharge increases (Koné et al., 2010; Bouillon et al., 2012), and is likely linked to the increased supply of
organic waste primed for decomposition from Nairobi. On the other hand, the highest peaks ($1857 - 2838$ nmol $L^{-1}$; $85171 -$
$135111\%$ saturation) were observed over the dry JJAS dry seasons of 2012 and 2013, their timing coinciding with the peaks
in POC, PN, and $NH_4^+$ previously discussed and attributed to large mammalian inputs, and we suggest these short-lived dry
season $CH_4$ peaks likely represent the decomposition of these mammalian-mediated terrestrial subsidies. $CH_4$ showed two
seasonal peaks, one during high water and another at the end of the low water period. The peak of $CH_4$ during high water might be
related to the increased connectivity between river and wetlands such as floodplains as reported in the Zambezi river (Teodoru et
al., 2015), and in the Oubangui (Bouillon et al. 2012; 2014). The peak of $CH_4$ at the end of the dry season is obviously unrelated to
interaction with wetlands since at this period river and floodplains are hydrologically disconnected. We hypothesize that this
increase of $CH_4$ is related to the combination of increase water residence time and the additional inputs of organic matter from
hippopotami. Indeed, they aggregate during low flow in river pools and river banks leading to a substantial input of organic matter
(Subalusky et al., 2015), that we hypothesise leads to enhanced in-stream $CH_4$ production. During high-water period, the
hippopotami disperse across the landscape, presumably having a lower impact on river water biogeochemistry. Indeed, during the
low water period $O_2$ decreased in 2011, although the $CH_4$ increase was modest. However, the marked increase of $CH_4$ at the end of
the 2013 dry season was mirrored by a distinct decrease of $O_2$ saturation level from ~$100\%$ to ~$20\%$. Although we provide no flux
estimates, these elevated concentrations relative to observations in other African river systems at least hint that the A-G-S
river system may be a relatively significant source of $CH_4$ outgassing at the local scale.
Nitrous oxide in rivers is sourced from either nitrification or denitrification, and although the interest in $N_2O$ is growing due
to its recognition as a significant contributor to radiative forcing (Hartmann et al., 2013) and as a major ozone depleting
substance (Ravishankara et al., 2009), relatively limited datasets are available for rivers (see Baulch et al. (2011); Beaulieu et



al. (2011); Marzadri et al. (2017)) and very few for tropical systems specifically (see Guérin et al. (2008); Bouillon et al.
(2012); Borges et al. (2015a)). We observe similar seasonal patterns in the Sabaki River as those observed by Bouillon et al.
(2012) in the Oubangui River, with concentrations during low flow conditions typically hovering between ~5 – 6 nmol L$^{-1}$
(Fig. 7b) and increasing as high flow conditions approach, though our peak concentration (26.6 nmol L$^{-1}$; 463% saturation)
is considerably higher than that reported for the largely pristine Oubangui River basin (9.6 nmol L$^{-1}$; 165% saturation), with
this pattern reflecting well the concentrations observed at the monitoring station during the basin-wide campaigns of JJAS
dry season (6.3 nmol L$^{-1}$; 116% saturation) and OND wet season (15.8 nmol L$^{-1}$; 274% saturation). The seasonal pattern
reported from these African rivers is unique compared to temperate rivers, where the opposite pattern is more typical (Cole
and Caraco, 2001b; Beaulieu et al., 2011). Given the reported correlation between $N_2O$ and $NO_3^-$ concentrations in various
river systems (Baulch et al., 2011; Beaulieu et al, 2011), including three from Africa (Borges et al., 2015a), and that basin-
wide data shows gradually increasing concentrations of $NO_3^-$ from ~179 μmol L$^{-1}$ to 538 μmol L$^{-1}$ over the 200 km reach
directly upstream of the monitoring site during the OND wet season (see Marwick et al. (2014a)), we make a first
assumption that the elevated $N_2O$ concentrations during the wet season may be driven by upstream nitrification of the
wastewater inputs identified in Marwick et al. (2014a).

## 4.3 Future outlook

The biogeochemical cycles and budgets of the Athi-Galana-Sabaki river system have been considerably perturbed by the
introduction of European agricultural practises in the early 20[th] century and the expanding population of Nairobi living with
inadequate waste water facilities (Van Katwijk et al., 1993; Fleitmann et al., 2007). These factors have had considerable
impact on riverine sediment loads (Fleitmann et al., 2007), instream nutrient cycling (Marwick et al., 2014a), and near-shore
marine ecosystems in the vicinity of the Sabaki outlet (Giesen and van de Kerkhof, 1984; Van Katwijk et al., 1993). Recent
modelling of nutrient export to the coastal zone of Africa to the year 2050 foreshadows continued perturbation to these
ecosystems, with the extent dependant on the land management pathway followed and mitigation strategies emplaced (Yasin
et al., 2010). Although suspended sediment fluxes are estimated to decrease over Africa in the coming 40 years, the
projected increase in dissolved forms of N and P and decreases in particulate forms of C, N, P as well as dissolved OC
(Yasin et al., 2010) will further augment nutrient stoichiometry within the inland waters of the A-G-S system.
Although no large reservoirs have been developed within the A-G-S basin, approval has been given for the construction of
the Thwake multi-purpose dam on the Athi River, though commencement has been delayed by tender approval for the
project. The total surface area is expected to be in the vicinity of 29 km$^2$, and the completed reservoir can be expected to
have a considerable impact on the downstream geomorphology and biogeochemistry of the river, as experienced in the
neighbouring reservoir-regulated Tana River (see Adams and Hughes (1986); Maingi and Marsh, 2002; Bouillon et al.
(2009); Tamooh et al. (2012), Tamooh et al. (2014); Okuku et al., (2016)). Given lakes and reservoirs enhance the cycling
and removal of nutrients due to their ability to prolong material residence times and subsequently enhance particle settling
and in-situ processing (Wetzel, 2001; Harrison et al., 2009), in addition to suggestions that GHG emissions from lentic





systems of the tropics may be disproportionately large relative to temperate and northern latitude systems (Aufdenkampe et
al., 2011; Bastviken et al., 2011; Raymond et al., 2013; Borges et al. 2015b), it is reasonable to assume the planned reservoir
on the Athi River will become a biogeochemical hotspot for the processing, storage and removal of upstream anthropogenic-
driven nutrient loads. The datasets presented within Marwick et al. (2014a) and here provide critical base-line data for future
research initiatives in the A-G-S system, not only to assess the evolving fluvial biogeochemistry of the basin in response to a
newly constructed tropical reservoir, but importantly, to review the influence flow regulation has on nutrient and suspended
sediment fluxes to the coastal zone, and subsequently the health and biodiversity of the Malindi-Watamu Marine National
Park ecosystem.

**Supplementary Materials**

Raw data and additional figures referred to in-text are included in the Supplementary Materials.

**Team list**

Trent R. Marwick
Fredrick Tamooh
Bernard Ogwoka
Alberto V. Borges
François Darchambeau
Steven Bouillon

**Author Contributions**

TRM: lead author, conceived research, performed field sampling, performed sample and data analysis, wrote paper. FT:
performed field sampling and sample analysis. BO: performed field sampling. AVB: conceived research, performed sample
analysis, wrote paper. FD: performed sample analysis. SB: conceived research, performed sample and data analysis, wrote
paper.

**Competing Interests**

The authors declare that they have no conflict of interest.

**Acknowledgements.**

Funding for this work was provided by the European Research Council (ERC-StG 240002, AFRIVAL,
http://ees.kuleuven.be/project/afrival/), and by the Research Foundation Flanders (FWO-Vlaanderen, project G.0651.09). We
thank Z. Kelemen (KU Leuven), P. Salaets (KU Leuven), and M.-V. Commarieu (ULg) for technical and laboratory
assistance, and J. Ngilu and W.R.M.A. (Water Resource Management Authority, Machakos, Kenya) for providing the
discharge data. Thanks also to Christopher Still and Rebecca Powell for providing the GIS data layers of their isoscape
models. A. V. Borges is a senior research associate at the FRS-FNRS (Belgium).

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
