# Peer review of "A comprehensive biogeochemical record and annual flux estimates for the Sabaki River (Kenya)"

_Biogeosciences, 2017_

## Referee Comment (RC1) · Anonymous Referee #1 · 7 Sep 2017

General comments:

This paper presents a two-year biogeochemical record (with biweekly sampling frequencies) of the Sabaki River and A-G-S river basin in Kenya. The authors seek to provide initial baseline data given the expected changes to the AGS river basin, such as: the growing contiguous population in Nairobi with inadequate sewage facilities, the anticipated increase in dissolved nutrient export from African river basins, and the planned damming of a river within the basin. While establishing this baseline is a critical need and the data collected for this effort is impressive in scope, the paper could greatly benefit from better organization around hypotheses and re-focusing based on

the data and statistical tools needed to test these hypotheses. Three general suggestions are highlighted below:

1. Hypothesis and purpose unclear; the comprehensive nature of the paper obscures the message. The authors are encouraged to identify a story (or stories) they can tell with these data and keep to that purpose. One potential action is to split into multiple papers if data support multiple, novel stand-alone documents. Once clear hypotheses are formulated, the paper should be trimmed to focus on the objective(s).

2. Novelty; the comprehensive past and current collaborations in the basin both benefit and detract from the strength of this manuscript. It is often unclear what is new vs. repackaged ($CH_4$, $N_2O$) vs. re-sampled (sediment fluxes, Marwick et al. 2014a) from previous publications, which muddies the novelty & distinct advancements made by this paper. The authors are encouraged to better highlight what is new.

3. Analysis/Statistical tests; the paper is lacking quantitative analyses and acknowledgements of uncertainty. Stats should follow hypotheses to test correlations between key parameters or multivariate models of interest. Time series analysis may be used to address time-varying correlations or controls on different biogeochemical fluxes. Further, if the authors wish to make quantitative comparisons of fluxes between different studies or solutes, they must acknowledge the uncertainty of their estimates (are the differences significant or just different within a similar range of uncertainty)?

Specific comments:

— Introduction - Starts out very C-focused. Overall - the introduction does not capture the objective(s) of the paper. Since a significant portion of the introduction is focused on C cycling/dynamics and metabolism, the reader is lead to believe that those topics will be a major focus of the manuscript. — Objectives of study - Not defined in the introduction, but clearly stated on pages 8 and 9, lines 12 and 1-3, respectively. Moving this section to the introduction (or re-wording it to fit into the introduction) would give the reader a much better understanding of the purpose. — What was the reasoning for

excluding CO2 data but including CH4 and N2O (e.g., Borges 2015a)? Failure to include CO2, DO, and metabolism data is a missed opportunity if this is to be a key focus of the paper. — Discharge data: gap-filling - the gaps and potential consequences of gaps in the discharge data must be addressed before making comparisons with other flux estimates. Given a 2 month period of no measurement - how off might the authors' estimates of missing Q from past years be? Have the authors tested the robustness of their gap-filling approach with other months that were not missing from the sampling period? — Discharge data: rating curve from gauge height - Discharge during much of the study period was well below and above the 2 clusters of points used to derive the rating curve (Fig 2a-b). What certainty do the authors have in making these interpolations and extrapolations from the 2 clusters and of the flux estimate comparisons that follow (e.g., Table 1, yields on p20)? — Nairobi - Referenced throughout text with little preface as to the location of the city in relation to the study area. "Nairobi" also appears to be used in place of urban influence (see page 5, line 15). Include clear explanation that Nairobi is the dominant "urban" influence in the study system. Introduction would benefit from additional literature supporting claim of anthropogenic influence on quantities of lateral nutrient inputs (if Nairobi or flow-regulated objectives become the primary) or whatever hypotheses the authors choose to test/focus on. — Many run-on sentences make key points difficult to follow (e.g., Page 3: 7-14, Page 21: 3-8, 8-14.)

Technical comments (noted by Page:Line):

Title: the title does not adequately capture the full scope of the paper. The title only mentions the Sabaki, yet the paper broadens its study site to the Athi-Galana-Sabaki (AGS) basin.

Abstract: the abstract was heavy in numbers, and read too much like a results section. It would benefit from more conceptual information.

2: 23: Consistency of "dammed" throughout is preferred (versus alternating with "flowregulated" when referring to dammed rivers).

3: 3-5: Here and throughout - try to stick to ~3 key references to make a point. Long lists are not helpful, and especially not needed if after "e.g.".

3: 5-7: Consider removing 'advancing to...global C cycle'

3: 9-11: Consider rewording "derived from heterotrophic metabolism…." in simpler terms

3: 21: Instead of "earth system domain", perhaps use biosphere?

3: 27: It is not clear why these regions would be more significant until later in the text. Reorganize and reorder.

6: 5: Figure "d" is the crop corrected vegetation, not "c"

7: 2: Here and throughout: don't need to define as physicochemical AND biogeochemistry unless the authors re-analyze data to include more processes or reactions (i.e., biogeochemistry). Otherwise delete biogeochemistry.

7: 4: What frequency were these temp, conductivity, O2, and pH data collected? These may be an interesting times series all to themselves….

7: 11: Was 2000mL of water collected at each sample or during the entire course of study?

8: 9: change 'period was provided' to 'period were provided'

8: 16-21: Perhaps this would be better suited in a discussion section than methods?

9: 15-16: Should the nutrient data collection time frame be mentioned here with discharge, or later with the nutrient data information?

11: Throughout section 3.3 - watch out for overuse of terms like "complex patterns", "complex variability", "no strong seasonal pattern", "erratic pattern", "complex variation", "highly variable", etc. These become overwhelming and the manuscript would be

much improved if they were removed and replaced with statistics.

12: 13: Here and throughout - consider re-arranging results to include fraction names with values instead of listing in separate clauses. For example: "4.0 Tg TSM yr-1, 70.6 Gg C-POC yr-1, and 24.1 Gg C-DOC yr-1."

19: 13: Opportunity to illustrate how a coarser sampling schedule may yield these differences in flux estimates: what would the authors conclude from this bi-weekly dataset if they trimmed it to the frequency of previous budget sampling intervals? Same difference or different results entirely?

21: While a very interesting side-note, this discussion using isotope values (but NOT mixing models or other quantitative tools) is a diversion from this paper as currently organized and seems better suited for a short note of its own.

Figure 1: Would be very helpful to show S19, S20, other key sampling sites - perhaps directly labeled in panel a.

Figure 6: Keep y-axis titles on the same side

---

## Referee Comment (RC2) · Anonymous Referee #2 · 25 Sep 2017

This is a well-structured and clearly written paper presenting a 2-year record of biogeochemical data from a drainage basin in Kenya. The paper is more constrained than the title suggests but the authors provide a full description of the trends observed and place this is the context of other studies. Given the focus of the paper, I feel that in its present form it is overlong, and would benefit from a more selective use of the literature: the introduction could be halved in length, with more emphasis on areas of novelty addressed by the paper, and providing clear aims / objectives. This is also scope to reduce the Discussion in length, but this should include a clearer statement of the significance of the work for readers. Overall I think the study is appropriate for publication in this journal, although in a revised paper, the authors might want to con-

sider: • Providing more detail on sampling protocols, processing and the timing of laboratory analyses; • Considering wider temporal trends (i.e. how representative are the two years of study); • Justifying the sampling location point – in the context of a large and heterogeneous catchment; • Reducing the number of studies cited – which seems excessive, given that the stated aim of the paper is 'to present a 2-year biogeochemical record'.
* * *

---

## Author Comment (AC1) · 14 Dec 2017

Below, we provide a point-by-point reply to the referee comments and suggestions, indicating if and how these were addressed in the revised version of our manuscript. We thank both reviewers for comments and suggestions that help us clarify the content of the manuscript.

Anonymous Referee #1

REF: General comments: This paper presents a two-year biogeochemical record (with biweekly sampling frequencies) of the Sabaki River and A-G-S river basin in Kenya.
The authors seek to provide initial baseline da ta given the expected changes to the AGS river basin, such as: the growing contiguous population in Nairobi with inadequate sewage facilities, the anticipated increase in dissolved nutrient export from African river basins, and the planned damming of a river within the basin. While establishing this baseline is a critical need and the data collected for this effort is impressive in scope, the paper could greatly benefit from better organization around hypotheses and re-focusing based on the data and statistical tools needed to test these hypotheses. Three general suggestions are highlighted below:

REF: 1. Hypothesis and purpose unclear; the comprehensive nature of the paper obscures the message. The authors are encouraged to identify a story (or stories) they can tell with these data and keep to that purpose. One potential action is to split into multiple papers if data support multiple, novel stand-alone documents. Once clear hypotheses are formulated, the paper should be trimmed to focus on the objective(s).

REPLY: While we understand that a lot of data are presented here, we do not feel they should be split up into multiple papers, this study was essentially a 2-year record of element fluxes and ancillary biogeochemical data from an understudied region, i.e. not particularly hypothesis-driven and we feel it would be better if the data collected stay together. In line with other suggestions below, we expect the various changes made in the abstract and introduction address this suggestion.

REF: 2. Novelty; the comprehensive past and current collaborations in the basin both benefit and detract from the strength of this manuscript. It is often unclear what is new vs. repackaged (CH4, N2O) vs. re-sampled (sediment fluxes, Marwick et al. 2014a) from previous publications, which muddies the novelty & distinct advancements made by this paper. The authors are encouraged to better highlight what is new.

REPLY: We have now indicated more clearly how this work relates to other studies from this basin.

REF: 3. Analysis/Statistical tests; the paper is lacking quantitative analyses and acknowledgements of uncertainty. Stats should follow hypotheses to test correlations between key parameters or multivariate models of interest. Time series analysis may be used to address time-varying correlations or controls on different biogeochemical fluxes. Further, if the authors wish to make quantitative comparisons of fluxes between different studies or solutes, they must acknowledge the uncertainty of their estimates (are the differences significant or just different within a similar range of uncertainty)?

REPLY: We fully understand this comment, but do not feel we have in hand to address this; this is the reason why we refrained from making quantitative comparisons with other flux studies except for general statements (e.g. in section 4.1, when comparing our sediment yields with earlier estimates). This is unfortunately a system where discharge data are scarce and not well constrained, hence our flux estimates should be considered as first-order estimates, as we feel should be evident as we show the rating curve and discuss its limitations. We do not feel we have made statements or tested hypothesis that require statistical tests.

REF: Specific comments:

REF: Introduction - Starts out very C-focused. Overall - the introduction does not capture the objective(s) of the paper. Since a significant portion of the introduction is focused on C cycling/dynamics and metabolism, the reader is lead to believe that those topics will be a major focus of the manuscript.

REPLY: None of the paragraphs in the introduction focuses on metabolism or processing of carbon. The intro sets the scene as to (i) why river systems are considered important in regional/global C budgets, (ii) the scarcity of basic datasets from numerous regions, (iii) the fact that these systems are undergoing rapid changes due to anthropogenic pressures.

REF: Objectives of study - Not defined in the introduction, but clearly stated on pages 8 and 9, lines 12 and 1-3, respectively. Moving this section to the introduction (or rewording it to fit into the introduction) would give the reader a much better understanding

of the purpose.

REPLY: The last paragraph of the introduction mentioned the objectives of our study: "Here, we present a 2-year biogeochemical record at fortnightly resolution for the riverine end-member of the A-G-S system, and in light of the planned construction of the Thwake Multi-purpose Dam (currently awaiting tender approval, see http://www.afdb.org/projects-and-operations/project-portfolio/project/p-ke-e00-008/), we provide estimates for sediment and nutrient export rates from the A-G-S system whilst still under pre-dam conditions." We have rephrased this to be more explicit and have also mentioned our objectives more clearly in the abstract.

REF: What was the reasoning for excluding CO2 data but including CH4 and N2O (e.g., Borges 2015a)? Failure to include CO2, DO, and metabolism data is a missed opportunity if this is to be a key focus of the paper.

REPLY: CO2 data are unavailable from the present data-set. All of the CO2 data reported by Borges et al. (2015a) were measured on-site during field expeditions but not from the monitoring at fixed stations. On the one hand, CO2 computed from pH and alkalinity is not always reliable (Abril et al. 2015) and maintaining high quality pH data throughout such a period at a remote site is complex, and on the other hand CO2 samples are not correctly preserved with HgCl2 due to precipitation of HgCO3. Hence, we recommend direct measurements of CO2 in the field with infra-red gas analysers, which was not possible in the present study.

REF: Discharge data: gap-filling - the gaps and potential consequences of gaps in the discharge data must be addressed before making comparisons with other flux estimates. Given a 2 month period of no measurement - how off might the authors' estimates of missing Q from past years be? Have the authors tested the robustness of their gap-filling approach with other months that were not missing from the sampling period?

REPLY: The 2-month data gap falls within the dry season, when flows do not vary much

and are consistently low. If this data gap would have fallen within the wet season, this would have been complicated to address reliably. We have now mentioned explicitly in the revised version of the ms that the data gap falls within the dry season.

REF: Discharge data: rating curve from gauge height - Discharge during much of the study period was well below and above the 2 clusters of points used to derive the rating curve (Fig 2a-b). What certainty do the authors have in making these interpolations and extrapolations from the 2 clusters and of the flux estimate comparisons that follow (e.g., Table 1, yields on p20)?

REPLY: We don't have the data needed to address this comment: to the best of our knowledge these are the only discharge measurements available and it is indeed unfortunate that they fall in two clusters and do not cover the full range of observed water heights. The only thing we can do (and did) is present these data in full transparency so that the readers are well aware of the data limitations. Nevertheless, note that the rating curve was fitted with an exponential function that is standard in hydrology and derived from theory (Kennedy 1984).

REF: Nairobi - Referenced throughout text with little preface as to the location of the city in relation to the study area. "Nairobi" also appears to be used in place of urban influence (see page 5, line 15). Include clear explanation that Nairobi is the dominant "urban" influence in the study system. Introduction would benefit from additional literature supporting claim of anthropogenic influence on quantities of lateral nutrient inputs (if Nairobi or flow-regulated objectives become the primary) or whatever hypotheses the authors choose to test/focus on.

REPLY: We have now indicated the location of Nairobi on Figure 1a, and mention this explicitly in the text.

REF: Many run-on sentences make key points difficult to follow (e.g., Page 3: 7-14, Page 21: 3-8, 8-14.)

REPLY: We have rewritten the sections referred to.

REF: Technical comments (noted by Page:Line):

Title: the title does not adequately capture the full scope of the paper. The title only mentions the Sabaki, yet the paper broadens its study site to the Athi-Galana-Sabaki (AGS) basin.

REPLY: We agree this may be rather confusing, the river is known as Athi upstream and as the Galana or Sabaki downstream of the confluence with the Tsavo River. Our sampling site was in the lower part of the river, i.e; on the Sabaki or Galana River. We have not modified the title as we do not want to suggest that we have flux data for different sites along the river, though we now clarify the nomenclature in the Materials and Methods section.

REF: Abstract: the abstract was heavy in numbers, and read too much like a results section. It would benefit from more conceptual information.

REPLY: We agree and have cut down the amount of numbers in the abstract.

REF: 2: 23: Consistency of "dammed" throughout is preferred (versus alternating with "flow regulated" when referring to dammed rivers).

REPLY: Amended as suggested by R1.

REF: 3: 3-5: Here and throughout - try to stick to 3 key references to make a point. Long lists are not helpful, and especially not needed if after "e.g.".

REPLY: Amended as suggested by R1.

REF: 3: 5-7: Consider removing 'advancing to...global C cycle'

REPLY: Amended as suggested by R1.

REF: 3: 11: Consider rewording "derived from heterotrophic metabolism: : ::" in simpler terms

REPLY: Amended to "...derived either from instream remineralisation of a proportion of lateral inputs, through inputs from groundwaters and floodwaters carrying the products of terrestrial mineralization (Cole and Caraco, 2001a; Beaulieu et al., 2011; Raymond et al., 2013),...".

REF: 3: 21: Instead of "earth system domain", perhaps use biosphere?

REPLY: Amended as suggested by R1.

REF: 3: 27: It is not clear why these regions would be more significant until later in the text. Reorganize and reorder.

REPLY: We re-organized this as suggested.

REF: 6: 5: Figure "d" is the crop corrected vegetation, not "c"

REPLY: Amended as suggested by R1.

REF: 7: 2: Here and throughout: don't need to define as physicochemical AND biogeochemistry unless the authors re-analyze data to include more processes or reactions (i.e., biogeochemistry). Otherwise delete biogeochemistry.

REPLY: Amended as suggested by R1.

REF: 7: 4: What frequency were these temp, conductivity, O2, and pH data collected? These may be an interesting times series all to themselves.

REPLY: These were discrete measurements carried out concurrently with the sample collection, i.e. at the same frequency.

REF: 7: 11: Was 2000mL of water collected at each sample or during the entire course of study?

REPLY: Amended for clarity to "Approximately 2000 mL of water was collected on each sampling occasion at ~0.5 m below the water surface...".

REF: 8: 9: change 'period was provided' to 'period were provided'

REPLY: Amended as suggested by R1.

REF: 8: 16-21: Perhaps this would be better suited in a discussion section than methods?

REPLY: We agree with the suggestion to move this, but since the Discussion did not have a section dedicated to the discharge, we moved it to the relevant section of the Results where this critical not is welcome.

REF: 9: 15-16: Should the nutrient data collection time frame be mentioned here with discharge, or later with the nutrient data information?

REPLY: Nutrient collection data time frame moved to precede presentation of nutrient results in '3.3 Bulk concentrations' section.

REF: 11: Throughout section 3.3 - watch out for overuse of terms like "complex patterns", "complex variability", "no strong seasonal pattern", "erratic pattern", "complex variation", "highly variable", etc. These become overwhelming and the manuscript would be much improved if they were removed and replaced with statistics.

REPLY: Valid point, we rephrased these terms in this section to be more consistent.

REF: 12: 13: Here and throughout - consider re-arranging results to include fraction names with values instead of listing in separate clauses. For example: "4.0 Tg TSM yr-1, 70.6 Gg C-POC yr-1, and 24.1 Gg C-DOC yr-1."

REPLY: Amended here and throughout as suggested by R1.

REF: 19: 13: Opportunity to illustrate how a coarser sampling schedule may yield these differences in flux estimates: what would the authors conclude from this bi-weekly dataset if they trimmed it to the frequency of previous budget sampling intervals? Same difference or different results entirely?

REPLY: This is a very good point, but again we do not feel we have the best dataset to address this. We recently did such an exercise for material fluxes in the Tana River

(Kenya) where we were fortunate to have much better data coverage and a more complete set of reliable discharge data (Geeraert et al. 2015 and Geeraert et al. under review). Obviously, the temporal resolution required to obtain robust estimates will depend on the flow variability.

REF: 21: While a very interesting side-note, this discussion using isotope values (but NOT mixing models or other quantitative tools) is a diversion from this paper as currently organized and seems better suited for a short note of its own.

REPLY: We understand this may seem a side track, but would prefer to keep this section in, since (i) here we can go beyond a simple quantitative (flux) study but include information on sources of carbon transported, which should be a relevant aspect of any riverine flux study, and (ii) we do not feel the isotope data are sufficiently extensive to make a separate paper.

REF: Figure 1: Would be very helpful to show S19, S20, other key sampling sites - perhaps directly labelled in panel a.

REPLY: Figure 1 has been modified to include the outline of Nairobi (cfr earlier comment) and key sampling sites.

REF: Figure 6: Keep y-axis titles on the same side.

REPLY: Figure 6 was modified as suggested.

References used in this reply:

Geeraert N, Omengo FO, Tamooh F, Paron P, Bouillon S, & Govers G (2015) Sediment yield of the lower Tana River, Kenya is insensitive to dam construction: sediment mobilization processes in a semi-arid tropical river system. Earth Surface Processes and Landforms, 40: 1827-1838.

Geeraert N, Omengo FO, Tamooh F, Marwick TR, Borges AV, Govers G, & Bouillon S (2017). Intra- and interannual variations in carbon fluxes in a tropical river system

(Tana River, Kenya). Under review.

Kennedy, E.J. (1984). Discharge ratings at gaging stations: U.S. Geological Survey Techniques of Water-Resources Investigations, Book 3. US Government Printing Office. https://pubs.usgs.gov/twri/twri3-a10/pdf/TWRI_3-A10.pdf 

Anonymous Referee #2

Below, we provide a point-by-point reply to the referee comments and suggestions, indicating if and how these were addressed in the revised version of our manuscript. We thank both reviewers for comments and suggestions that help us clarify the content of the manuscript.

REF: This is a well-structured and clearly written paper presenting a 2-year record of biogeochemical data from a drainage basin in Kenya. The paper is more constrained than the title suggests but the authors provide a full description of the trends observed and place this is the context of other studies. Given the focus of the paper, I feel that in its present form it is overlong, and would benefit from a more selective use of the literature: the introduction could be halved in length, with more emphasis on areas of novelty addressed by the paper, and providing clear aims / objectives. This is also scope to reduce the Discussion in length, but this should include a clearer statement of the significance of the work for readers. Overall I think the study is appropriate for publication in this journal, although in a revised paper, the authors might want to consider:

REF: i. Providing more detail on sampling protocols, processing and the timing of laboratory analyses;

REPLY: We have provided a few minor details on methodology of sampling and sample processing, but do not really see where we could expand without getting lost in detail.

REF: ii. Considering wider temporal trends (i.e. how representative are the two years of study);

REPLY: Valid question, but we do not feel we have the data to say something meaningful, we have now included a sentence mentioning that obviously, our estimates are only a snapshot and that one could expect strong inter-annual variability typical of semi-arid rivers (e.g. Geeraert et al. 2015 for the nearby Tana River).

REF: iii. Justifying the sampling location point – in the context of a large and heterogeneous catchment;

REPLY: We have added a short statement justifying the location of the sampling site.

REF: iv. Reducing the number of studies cited – which seems excessive, given that the stated aim of the paper is 'to present a 2-year biogeochemical record'.

REPLY: We fully agree; a similar suggestion was made by Ref#1, and we have cut down the number of references for many statements considerably.

References used in this reply:

Geeraert N, Omengo FO, Tamooh F, Paron P, Bouillon S, & Govers G (2015) Sediment yield of the lower Tana River, Kenya is insensitive to dam construction: sediment mobilization processes in a semi-arid tropical river system. Earth Surface Processes and Landforms, 40: 1827-1838